# On the statistical properties of sea ice lead fraction and heat fluxes in the Arctic

Einar Ólason[1], Pierre Rampal[1,2], and Véronique Dansereau[1,3]

[1]Nansen Environmental Remote Sensing Center and Bjerknes Centre for Climate Research, Bergen, Norway
[2]Now at Université Grenoble Alpes, CNRS, Grenoble INP, Institut de Géophysique de l'Environnement, Grenoble, France
[3]Now at Université Grenoble Alpes, CNRS, Grenoble INP, Laboratoire 3SR, Grenoble, France

**Correspondence:** Einar Ólason (einar.olason@nersc.no)

**Abstract.** We explore several statistical properties of the observed and simulated Arctic sea-ice lead-fraction, as well as the statistics of simulated Arctic ocean-atmosphere heat fluxes. First we show that the observed lead fraction in the Central Arctic has a monofractal spatial scaling, which we relate to the multifractal spatial scaling present in sea-ice deformation-rates. We then show that the relevant statistics of the observed lead fraction in the Central Arctic are both well represented by our model, neXtSIM. Given that the heat flux through leads may be up to two orders of magnitude larger than that through unbroken ice we then explore the statistical properties (PDF and spatial scaling) of the heat fluxes simulated by neXtSIM. We demonstrate that the modelled heat fluxes present a multifractal scaling in the Central Arctic, where heat fluxes through leads dominate the high-flux tail of the PDF. This multifractal character relates to the multi- and monofractal character of deformation rates and lead fraction. In the wider Arctic, the high-flux tail of the PDF is dominated by an exponential decay, which we attribute to the presence of coastal polynyas. Finally, we show that the scaling of simulated lead fraction and heat fluxes depend weakly on the model resolution and discuss the role sub-grid scale parameterisations of the ice heterogeneity may have in improving this result.

## 1 Introduction

Sea ice is well known to be an excellent insulator. In the Arctic, it reduces the potential flux of heat from the ocean to the atmosphere by two orders of magnitude in the winter (e.g., Maykut, 1986; Andreas et al., 1979). Expanses of open water such as leads and polynyas on the other hand act to release substantial amounts of heat and moisture into the atmosphere, thereby promoting the production of new ice and the rejection of brine into the ocean. In particular, coastal and flaw polynyas produce large amounts of ice and their role in Arctic climate and oceanography has been widely studied (e.g., Aagaard et al., 1981; Winsor and Björk, 2000; Morales Maqueda et al., 2004; Tamura and Ohshima, 2011). Leads, on the other hand, are much more difficult to study because they are much narrower and shorter lived than polynyas, and at the same time can form anywhere in

the Arctic Basin. However, processes surrounding lead formation are of considerable local importance and may have significant climatic influence as well, despite their extreme localisation (see for example Lüpkes et al., 2008a, b; Vihma et al., 2013, for an overview).

When a lead opens in the ice during winter, relatively warm ocean waters become exposed to the cold atmosphere, resulting in heat fluxes of up to 600 W/m$^2$ (Maykut, 1986; Andreas and Murphy, 1986). As a result, a plume of warm, moist air forms above the lead, driving convection in the predominantly stable or near-neutral Arctic atmospheric boundary layer (ABL). In the presence of a capping inversion, the plume may penetrate the lowest levels of the inversion as it rises, leading to entrainment. Using data from an aircraft campaign, Tetzlaff et al. (2015) identified such entrainment for all scenarios they encountered, with entrainment fluxes reaching up to 30% of the surface heat fluxes. They also found clear evidence that this entrainment contributed significantly to warming the ABL downwind of the lead.

The transport of sensible heat flux, moreover, is significant over leads and was shown to be even more efficient over smaller than larger leads (Andreas and Cash, 1999; Esau, 2007). Following up on the work of Andreas and Cash (1999), Marcq and Weiss (2012) estimated the lead fraction from a SPOT satellite image, covering $60 \times 66$ km$^2$. They demonstrated that turbulent heat transfers between the ocean and the atmosphere in ice-covered oceans strongly depends on the distribution of lead widths, especially at very small scales (smaller than 50 m). This points to a potentially significant misrepresentation of heat fluxes in large-scale atmospheric and coupled models. Traditional sea-ice models are well known to only reproduce lead-fraction and linear-kinematic-feature properties when run at very high resolution and/or when care is taken that the momentum equation solver converges Girard et al. (2011); Wang et al. (2016); Spreen et al. (2017); Hutter et al. (2018). Neither of these requirements are met for the vast majority of model simulations used to study ice–ocean–atmosphere interactions in the Arctic, as the computational cost of doing so is substantial, although some progress is being made in this respect (Koldunov et al., 2019).

On the ocean side, the formation of new ice in leads removes fresh water and releases brine. Data from various field campaigns (e.g., Smith, 1974; Morison et al., 1992; Morison and McPhee, 1998), as well as numerical model results (e.g., Kozo, 1983; Smith and Morison, 1993; Smith et al., 2002) give a very consistent picture of brine release in leads. When the ice velocity is small or moderate, salt plumes form below the lead and sink to the bottom of the mixed layer. The plumes cannot penetrate the halocline but instead spread horizontally along the top of the halocline, reducing the depth of the mixed layer. When the ice velocity is sufficiently large, turbulent mixing occurs along the edges of the lead that distributes the rejected brine throughout the mixed layer. This process leads to large-scale convection in the mixed layer which deepens it and causes a vertically uniform salinity increase. Nguyen et al. (2009) showed that a realistic simulation of the Arctic halocline depends on the proper simulation of brine release and its redistribution in the water column, while Barthélemy et al. (2015) demonstrated the importance of representing both the high- and low-velocity regimes when parameterising brine release.

Leads, therefore, have a potentially significant role to play in the Arctic, through both their impact on the local weather conditions and their long-term influence on the state of the atmosphere and ocean. Even though their role in the ocean–atmosphere interaction is being actively researched (e.g., Esau, 2007; Lüpkes et al., 2008a, b; Marcq and Weiss, 2012; Chechin et al., 2019; Li et al., 2020), and the mechanisms of lead formation are well known, large-scale sea-ice models have not yet been

shown to robustly reproduce the statistical properties of lead fraction — as large-scale models cannot simulate single leads, only the frequency of their occurrence within a grid cell. When sea ice deforms, ridges and leads form. Consequently, the probability distribution functions (PDF) of open water densities, floe sizes, and deformation rates share the common property of a "heavy tail" (Rothrock and Thorndike, 1984; Matsushita, 1985; Stern et al., 2018; Weiss, 2003; Marsan et al., 2004; Marcq and Weiss, 2012) that is a signature of scale-invariance. Deformation rates (Marsan et al., 2004; Hutchings et al., 2011; Bouillon and Rampal, 2015a; Rampal et al., 2019) and open water densities (Weiss and Marsan, 2004) have been shown to display multifractality in the space domain and, in the case of deformation rates, in the time domain also (Weiss and Dansereau, 2017; Rampal et al., 2019).

The fractal characteristics of deformation rates and other quantities are of particular interest because their presence may provide interesting information about the underlying mechanisms at play in the physical system. In the case of deformation of geophysical solids, such as as sea ice, a fractal characteristic comes about, at least in part, because of a propagation of fracturing events, which can be modelled by multiplicative cascades (Weiss and Marsan, 2004). Fracturing triggers a redistribution of stresses in the ice, which in turn may give rise to further ice deformation nearby. Large deformation events are also likely to be recurrent where previous fracturing has already weakened the ice, resulting in a multi-fractal character (Weiss and Marsan, 2004; Marsan and Weiss, 2010). A well known analogue is crustal deformation, where earthquakes cause a redistribution of stresses in the Earth's crust, with large quakes clustered around weak areas in the crust and smaller quakes occurring in the wider vicinity (Kagan, 1991). In this study, we investigate how the fractal character of sea-ice deformation (see Bouillon and Rampal, 2015a; Rampal et al., 2016, 2019) affects lead fraction and heat fluxes through the ice, using both, observations and the neXtSIM model.

In Section 2 we briefly present the model setup, as well as the data and the methodology of the scaling analysis performed in this study. In Section 3 we present the probability density function (PDF) and spatial scaling of the observed lead fraction. We also demonstrate that the PDF and spatial scaling simulated with neXtSIM match well with those observed. The capability of the model to reproduce lead fraction statistics justifies its use in further analysing atmospheric heat flux statistics and inferring the role of leads in determining the properties of these statistics (Section 4). Following this we briefly investigate the influence of model resolution on our results in Section 5. There we show that the lead-fraction scaling and the heat-flux scaling depend only weakly on the model resolution. In Section 6 we then discuss the model evaluation against observations and the origin of the shape of the PDF of heat fluxes and their spatial scaling, for both the so-called "Central Arctic" (i.e., excluding coastal areas) and the whole Arctic Basin. We also discuss the source of the multifractal scaling for the heat fluxes and the role of surface heterogeneity in the localisation of heat fluxes at different model resolutions.

## 2 Model, data, and methodology

### 2.1 Model set-up

We use the latest version of the next generation sea ice model, neXtSIM, presented in Rampal et al. (2019). NeXtSIM is a stand-alone sea-ice model, coupled to a slab ocean and forced by the results of atmospheric and oceanic reanalyses. It uses the

Maxwell-Elasto-Brittle (MEB) rheology of Dansereau et al. (2016), a Lagrangian moving mesh as described in Rampal et al. (2016) and the two-layers thermodynamic model of Winton (2000). Heat fluxes between the ocean, ice, and atmosphere are calculated using traditional bulk formulae, as outlined by Rampal et al. (2016). Oceanic heat loss results in lowering of the slab ocean temperature, which may be compensated for by new-ice formation and nudging of the slab ocean layer temperature to reanalysis results. The model has in essence three ice categories: those of thick ice, open water, and newly formed thin ice. The ice thickness redistribution and thermodynamic schemes are outlined in the appendix of Rampal et al. (2019). All output variables are interpolated using a conservative scheme from the moving Lagrangian model mesh onto a fixed and regular Eulerian grid and are averaged on a daily basis to match the temporal resolution of the observations (see Section 2.2).

The model set-up covers the central Arctic Ocean, with open boundaries at the Bering Strait, the Canadian Arctic Archipelago, Greenland, and the Barents and Kara seas (see Fig. 1). The model's triangular mesh is built on the 6.25 km resolution grid of the lead fraction data set (Ivanova et al., 2016, see section 3) such that the two have the same coast lines and comparable resolutions. This is done to simplify the comparison of the two. In all other respects the model setup is the same as that of Rampal et al. (2019): it is forced using the ocean state from the TOPAZ4 oceanic reanalysis (Sakov et al., 2012) and the atmospheric state of the Arctic System Reanalysis[1] (Bromwich et al., 2016). The model is initialised with sea ice concentration and thickness from the OSISAF climate data record (Tonboe et al., 2016) and ICESAT[2] (Kwok et al., 2009) datasets respectively. We use results from the TOPAZ4 reanalysis to fill in gaps in the ICESAT thickness. The initial snow thickness is set based on the Warren et al. (1999) climatology and ice age, using half the climatological snow thickness over first year ice. We start the model run on November 15th, 2006, restricting our analysis to the winter months of January, February and March (JFM), 2007, so as not to influence the results by the very different heat fluxes and lead fractions seen during the freeze-up and melt periods.

We run three simulations, in addition to the control simulation, covering the same space domain and time period. The forcings are the same in the three cases. Two of these simulations investigate the effect of changing the model spatial resolution, and one investigates the effect of changing the model's rheological framework. We run the simulations related to model resolution at 12.5 and 25 km resolution, with all model parameters kept the same as in the control run. An exception to this is the cohesion parameter of the Mohr-Coulomb criterion, $c$, as this parameter scales with the model resolution as $c \sim 1/\sqrt{L_m}$, where $L_m$ is the model resolution (see Bouillon and Rampal, 2015a, for further details).

In the third simulation, the MEB rheology is replaced with a linear viscous rheology. The deficiencies of the linear viscous model are well known, and it is neither suited for a detailed study of the model physics nor model evaluation (see e.g., Leppäranta, 2005, and references therein). It is used here to investigate, in a simple and straightforward manner, the effect of not simulating highly localised leads in the Arctic, while at the same time simulating some polynya formation. As the solution of the linear viscous model quickly degrades we initialise the model with smoothed model results from the control run at weekly intervals, giving the model three days to spin up after each initialisation. This way the model solution has some time to

---

[1]https://rda.ucar.edu/datasets/ds631.0, ASRv1 30-km, formerly ASR final version, Byrd Polar Research Centre/The Ohio State University. Accessed 15 April 2015

[2]https://icdc.cen.uni-hamburg.de/1/daten/cryosphere/seaicethickness-satobs-arc.html

evolve after initialisation, but not enough time to diverge significantly. The value we use for the viscosity parameter coincides with the lower bound suggested by Hibler (1979) ($\zeta = 1.0 \times 10^{10}$ kg/s), as this gives a reasonably good drift speed compared to the OSI-SAF drift product, in our set-up. We did not attempt to tune the viscosity value further, as this model run proved sufficiently adjusted for the purposes of this study.

## 2.2 Observational data

We analyse observed lead fraction from Ivanova et al. (2016), as well as using it to evaluate the model results. This product improves on the original product from Bröhan and Kaleschke (2014) by correcting an overestimation of the lead fraction by a simple adjustment of the upper tie point used in the method. This product is based on passive microwave observations of the AMSR-E and is a daily, light and cloud independent, pan-Arctic data set, available from November to April, for the period 2002–2011. The dataset resolution is 6.25 km and the method allows the detection of leads wider than 3 km, meaning that a substantial amount of smaller leads are undetected in this product.

The data show the area fraction of each grid cell that is covered by leads filled with open water, thin ice, or a mixture thereof. The observations are filtered for feature orientation and the product, therefore, shows only the fraction of leads, excluding larger, non-linear features such as seasonally thin ice and polynyas. Although the thickness threshold for thin-ice detection in this product is not known precisely, it is unlikely that this approach classifies ice thicker than about 0.1 m as thin ice (Röhrs and Kaleschke, 2012).

## 2.3 Methodology

We briefly outline the methodology for investigating the statistical properties of lead fraction and heat fluxes here. More details can be found in Schertzer and Lovejoy (1987), Marsan et al. (2004), and Rampal et al. (2019). The first step in characterising the statistics of a process that exhibits scale-invariance is to consider the PDF of its realisations or event magnitudes. If the PDF has a tail that is "fat" (e.g., a power law, stretched exponential, log-normal), then there is potential for the investigated variable to be subject to fractal scaling. The slope of such a tail indicates the extent to which extreme events dominate the process studied and which moment orders are required to properly describe the distribution of event magnitudes. The PDFs shown in this paper are calculated from all the daily means for JFM, 2007, within the regions outlined below (Section 3), for both the lead fraction and heat flux magnitudes.

The second step is to investigate changes in the PDF of both the lead fraction and heat flux magnitudes (hereafter referred to as heat fluxes) with respect to the scale of observation. In this study, we focus on the space domain and, therefore, fix the temporal scale of analysis to one day. We choose the daily time scale to consistently compare simulated and observed lead fraction, and we retain the daily time scale for the sake of simplicity when analysing the heat fluxes. The scaling analysis consists of evaluating the different moments of the distribution at different spatial scales. The moments of the distribution are calculated as

$$\mu_q = \frac{1}{N} \sum_{j=1}^{N} x_j^q, \tag{1}$$

where $q$ is the moment order, $N$ is the number of samples, and $x_j$ are individual samples.

In the coarse-graining analysis, the mean and higher-order moments are calculated across a range of spatial scales, $L$, by averaging the observed and simulated values onto incrementally coarser grids. In order to calculate the mean scaling for JFM, we calculate the scaling for each daily mean and take the temporal mean of the means and moments calculated at each spatial scale. In the Eulerian case, the coarse-graining grid is set by the averaging grid, while in the Lagrangian case the coarse-graining grid can be chosen arbitrarily. In the Lagrangian case, we follow Marsan et al. (2004) and combine the results of differently placed coarse-graining grids at each spatial scale to improve the robustness of our statistics. For spatial scales larger than that of the observations or model, each cell of the coarse-graining grid may consist of a large number of land points (in the Eulerian/lead-fraction case) or few model elements (in the Lagrangian/heat-flux case). In the Eulerian case, we therefore assume that if the number of land points is more than half the points in a cell of the coarse-graining grid the data in that cell is not reliable and we discard it. In the Lagrangian case, we discard data if the number of model elements in a coarse-graining cell is less than half the median number of elements in all the cells.

Using the spatial coarse-graining method we we derive the moment values as a function of the scale. When presented in log-log space, the moment values should decrease linearly with increasing $L$. The slope of this linear decrease, $\beta$, is estimated for each moment order, $q$. The slope expresses a spatial scaling such that the moments are $\mu_q \sim L^{\beta(q)}$ and the mean is $\bar{x} \sim L^{-\beta(1)}$, where $L$ is the spatial scale. The structure function, $\beta(q)$, describes the change in the slope of the scaling as a function of the moment order. We estimate the uncertainty relative to this calculation as the 95% confidence interval of a least-squares linear fit of the $(L, \bar{x}(L))$ and $(L, \mu_q(L))$ points.

For a quantity related to a so-called scale invariant process, there is generally a monotonic change in $\beta$ with increasing $q$. The scaling is monofractal if $\beta(q)$ is linear and multifractal if $\beta(q)$ is parabolic. If the scaling is monofractal only the spatial organisation of the quantity is following a fractal pattern, with no dependence to the actual magnitude. If it is multifractal, however, the higher values are also distributed following a fractal pattern and thus are more localised than lower values.

## 3   Model evaluation against observed lead fraction

In this section, we demonstrate the capability of neXtSIM to reproduce lead-fraction statistics by comparing the statistical properties of simulated and observed lead fractions. As the observed lead fraction corresponds to the fraction of open water as well as thin ice (Röhrs and Kaleschke, 2012), we define the simulated lead fraction for the purpose of this comparison as the sum of the simulated open water fraction and of the fraction of new ice that is thinner than a given threshold. The correct threshold for thin ice is not well constrained since the maximum ice thickness that is classified as a lead in the satellite data is not well defined either. We, therefore, choose a threshold thickness for the model that the same slope of the PDF as the observed one, as shown below. For the JFM average, this optimal threshold is 0.098 m, but variations of this value by about 0.01 cm still give good agreement with the observations. This value is reasonable, given that Röhrs and Kaleschke (2012) estimates an upper bound on thin ice at 10 m in their product.

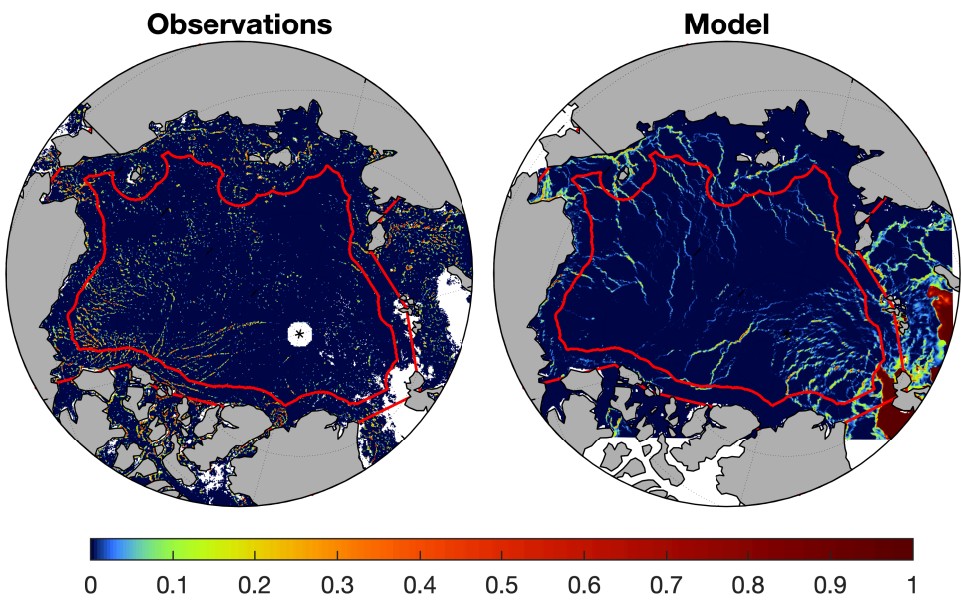

**Figure 1.** Observed and simulated daily mean lead fraction on March 14th, 2007. The figure shows the entire model domain, and the red lines indicate the boundaries of the "Arctic" (outer region) and "Central Arctic" regions used in the study.

It is important to note that the simulated lead fraction is not strictly a lead fraction as it includes all open water areas, including polynyas (cf Fig. 1). In contrast, the observed lead fraction data is filtered so that polynyas are left out of the final product. To allow for a fair comparison of the simulated and observed lead fraction, we therefore define a polynya-free region referred to here as the "Central Arctic", which covers the area more than 400 km northward of the 20 m isobath (see Fig. 1).

Figure 2 shows the PDF of observed and simulated lead fraction on a log-log scale. On this scale, both the observed and simulated PDF shows a linear decrease for high lead fraction, as expected based on the work of Marcq and Weiss (2012). The threshold for this behaviour is at about 40%, which roughly coincides with the sensitivity limit of the satellite product. For lead fractions larger than about 40% the slopes of the PDF for the observed and modelled lead fraction are very similar, when the thin ice threshold is chosen correctly. The choice of thin ice threshold for the model is based on matching these slopes ($-3.9$) over the lead fraction range of $[0.40, 0.93]$. For values smaller than about 40%, the slope of the PDF of the observations changes to approach zero for very small values. This behaviour is to be expected as the small leads are known not to be captured by the AMSR-E because of its resolution limitation and, therefore, are not present in this product. It is worth noting that although the spatial resolution of the model is similar to the resolution of the satellite product, the model can capture smaller leads,

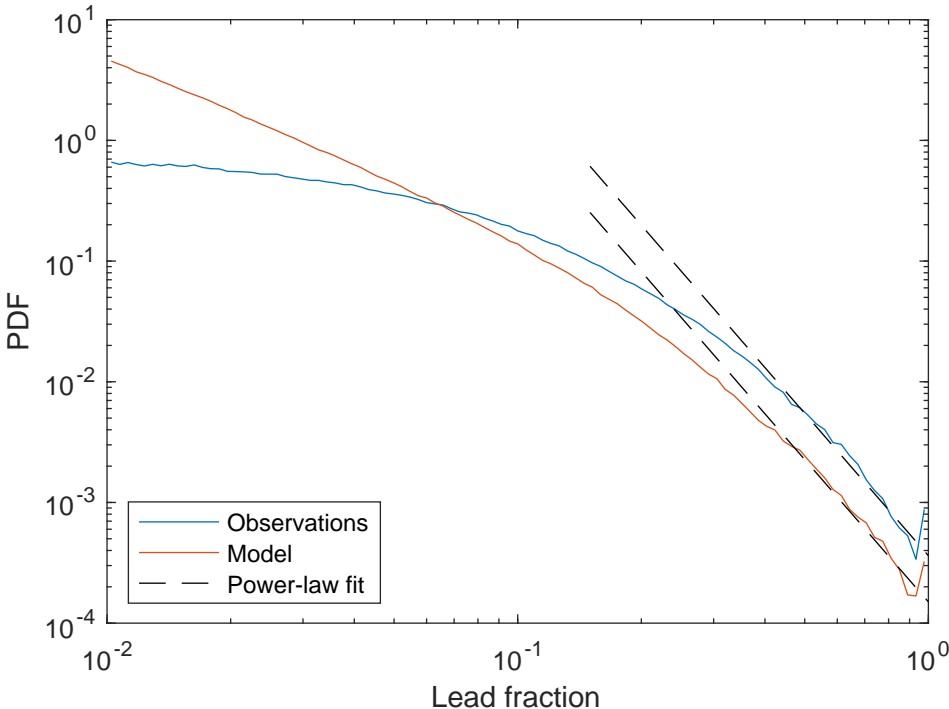

**Figure 2.** The probability density function of observed and simulated lead fraction in the "Central Arcitic" areas over JFM 2007. The dashed black lines are linear fits discussed in the text.

demonstrated by the absence of flattening of the PDF for small values. The slope of the simulated PDF does change at smaller values, becoming linear again below 2% (not shown).

We now consider the spatial scaling of the lead fraction for the first four moments of the distribution, as the absolute value of the slope of the PDF in the lead fraction range of $[0.40, 0.93]$ lies between 4 and 5 (Marsan et al., 2004). The spatial scaling, along with the resulting structure function, is given in Fig. 3 and shows a good agreement between the model and observations. The mean is conserved across scales, as expected, but the mean lead fraction is higher in the model than the observations. We know that small leads are not detected in the satellite product and we should, therefore, only compare model and observations in the high lead-fraction range, where we know the observations to be reliable. The mean observed and simulated lead fractions are $0.0055$ and $0.0047$. However, the means of the observed and simulated lead fractions larger than 40% are $0.554$ and $0.552$ respectively. The higher order moments of the observed and simulated lead fractions are in good agreement for all lead fraction values, but are virtually identical if we consider only lead fractions larger than 40%. This shows that the agreement between model and reality may be much better than a first-order interpretation of Fig. 3 would suggest. We suggest that a fairer comparison between the model simulation and observations would, therefore, consider only lead fractions larger than a

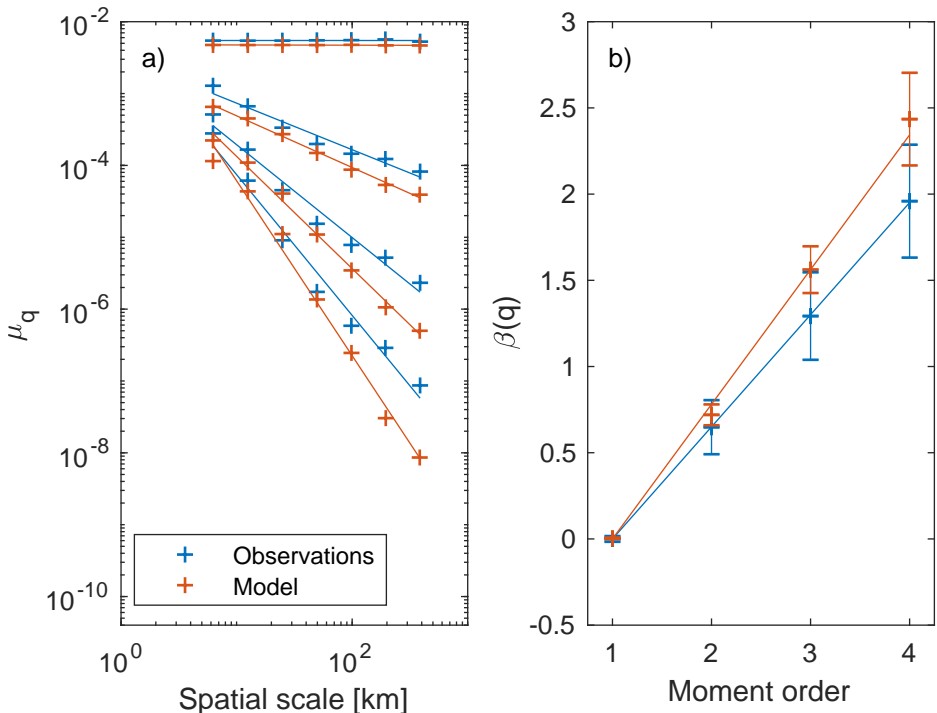

**Figure 3.** Left pane: The spatial scaling of observed and simulated lead fraction in the "Central Arctic" region over JFM 2007. The lines are fits for each moment (order 1, 2, 3, and 4 from top to bottom). Right pane: The resulting structure function for both observed and simulated lead fraction. The lines are linear least-squares fits and the error bars are the uncertainty of the least-squares fit.

given fraction. We have here used 40%, but it is not immediately obvious what the appropriate threshold fraction would be. We expect this approach to also greatly complicate the spatial scaling calculations.

The structure function underlines the good agreement between the simulation and observations: within the estimated uncertainties, the slopes of the observed and simulated structure functions are $0.650 \pm 0.006$ and $0.78 \pm 0.06$ respectively using a least-squares fit (see Fig. 3). We note that this structure function is linear and therefore that the scaling is monofractal. The good agreement between the observed and modelled structure function, together with the good agreement between observed and modelled mean and higher order moments accounting for only lead fractions larger than 40% (see above) is a strong

indicator that the model is simulating lead formation in a physical and realistic manner — even if the incompleteness of the observations does not allow a closer comparison than that done here.

## 4    Modelled heat fluxes

We now explore the statistical properties of the heat fluxes simulated by neXtSIM. Figure 4 shows the PDF of simulated heat fluxes for both the "Arctic" and "Central Arctic" regions. When plotted on a log-log scale the PDF shows a clear linear tail

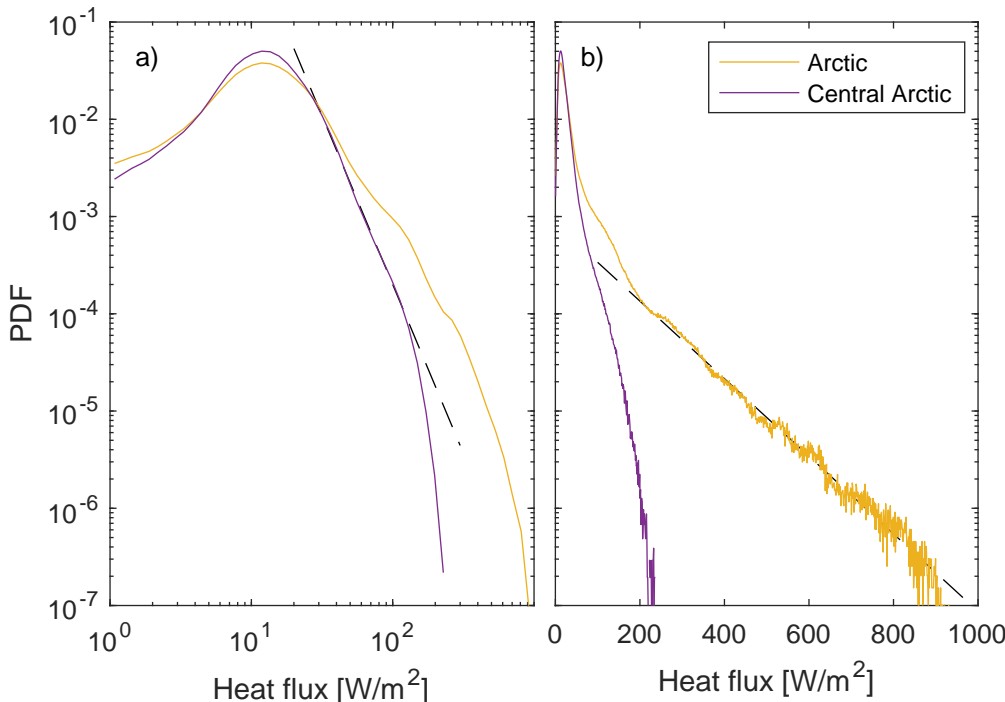

**Figure 4.** The probability density function of modelled atmospheric heat flux in the "Arctic" and "Central Arctic" regions over JFM. The dashed lines are linear fits discussed in the text.

for heat flux values between 30 and 110 W/m$^2$, with a slope of $-3.4$, within the Central Arctic region (Fig. 4a), dropping off rapidly for larger flux values. In the Arctic region, however, the tail is not linear on the log-log plot, but it is linear on a semi-log plot for heat flux values larger than 200 W/m$^2$ (Fig. 4b). We, therefore, expect to see spatial scaling only within the Central Arctic region. We note that the exponential decay displayed in the Arctic region is most likely related to the large coastal and flaw polynyas impacting the heat fluxes there.

Given the results of the analysis of the PDF, we consider the first three moments for the spatial scaling of heat fluxes, confining the analysis to the Central Arctic region. The results of this analysis are plotted in Fig. 5, which shows a clear scaling of the heat fluxes in space. Although there is substantially more scatter in this plot than in the lead fraction analysis (Fig. 3) it still clearly indicates that the scaling is multifractal.

     To demonstrate the role of leads in the scaling obtained for the simulated heat fluxes we ran the model with a linear viscous

rheology, as described in section 2.1. In this case, no leads form in the Central Arctic, but coastal and flaw polynyas do form. The PDF of the heat fluxes simulated with the viscous model over JFM 2007 is shown in Fig. 6. This experiment demonstrates that when using the linear viscous model, the tails of the heat-flux distribution are significantly reduced in the Central Arctic

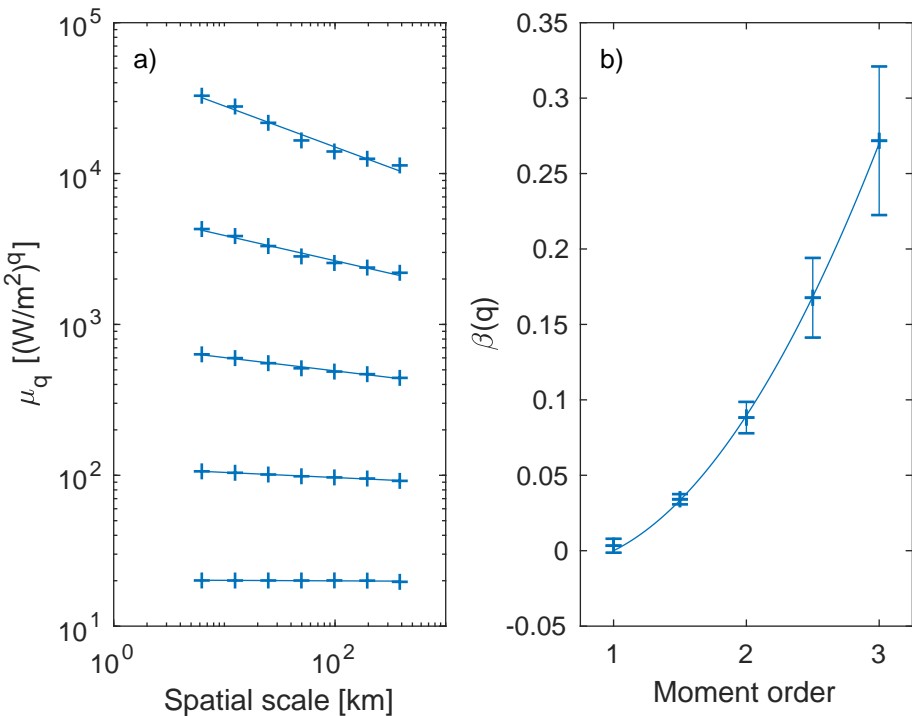

**Figure 5.** Left pane: The spatial scaling of simulated heat fluxes in the "Central Arctic" region over JFM 2007. The lines are linear least-square fits for each moment (order 1, 1.5, 2, 2.5, and 3 from bottom to top). Right pane: The resulting structure function and a quadratic fit. The error bars are the uncertainty of the least-squares fit for the moments (left pane).

region, so much so that one should suspect the absence of spatial scaling there. A spatial scaling analysis region indeed confirms that both the mean and the higher moments are the same at all spatial scales (not shown), as expected for a homogeneous fluid.

In the Arctic region we still obtain an exponential decay with the viscous model. This result supports the idea that this exponential function is the expression of the presence of polynyas in that region, since polynyas are present in both the MEB and viscous simulations.

## 5   Effects of resolution

To briefly explore the effects of the model resolution on the statistics of simulated lead fraction and heat fluxes we ran the model

in the same configuration, but at 12.5 km and 25 km resolution (see section 2.1). The comparison shows that the simulated lead-fraction scaling depends on model resolution, as all three model runs give different mean and higher order moments, and structure function (Fig. 7. The difference in the mean for the chosen resolutions is 30% between the 6.25 and 12.5 km resolution runs and 20% between the 12.5 and 25 km resolution runs, while higher order moments see increasing differences towards smaller spatial scales.

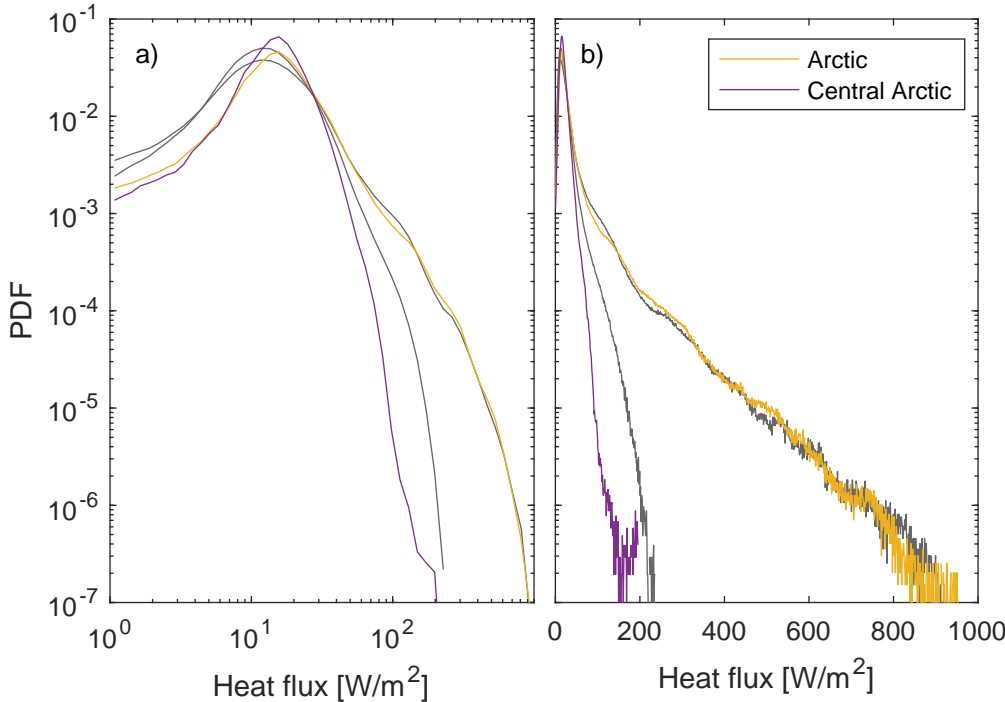

**Figure 6.** The probability density functions of modelled atmospheric heat flux in the "Arctic" and "Central Arctic" regions over JFM, using the viscous (coloured lines) and MEB (grey lines) rheological models.

This result means that running the model at different resolutions will give similar values for the mean, while for the higher order moments model runs at different resolutions will give different values. The difference between the runs at 6.25 and 12.5 km resolution are only modest, while going to 25 km resolution results in significant differences for the third and fourth moments, with a much less cleaner scaling as well. These differences also result in a change in the structure function, which could be multifractal, rather than monofractal for the 25 km resolution. The significance of this is limited by the large uncertainties

associated with the scaling at 25 km resolution.

For the simulated heat fluxes there are similarities but also clear differences between the results of simulations at different model resolutions (Fig. 8). The mean heat flux is nearly the same for all simulations and conserved at all scales. All three model realisations also show a clear scaling of the higher order moments and very similar statistics at the lowest spatial scales. The main difference between the model results is that, at small spatial scales, higher model resolution gives higher values for

the higher order moments. The slopes of the scaling and, therefore, the values of the structure function, are thus higher for the higher resolution runs.

In addition to these differences in the scaling, there also seems to be a difference in the nature of the structure function, depending on the model resolution. The structure function at 6.25 km resolution clearly indicates a multifractal scaling. This

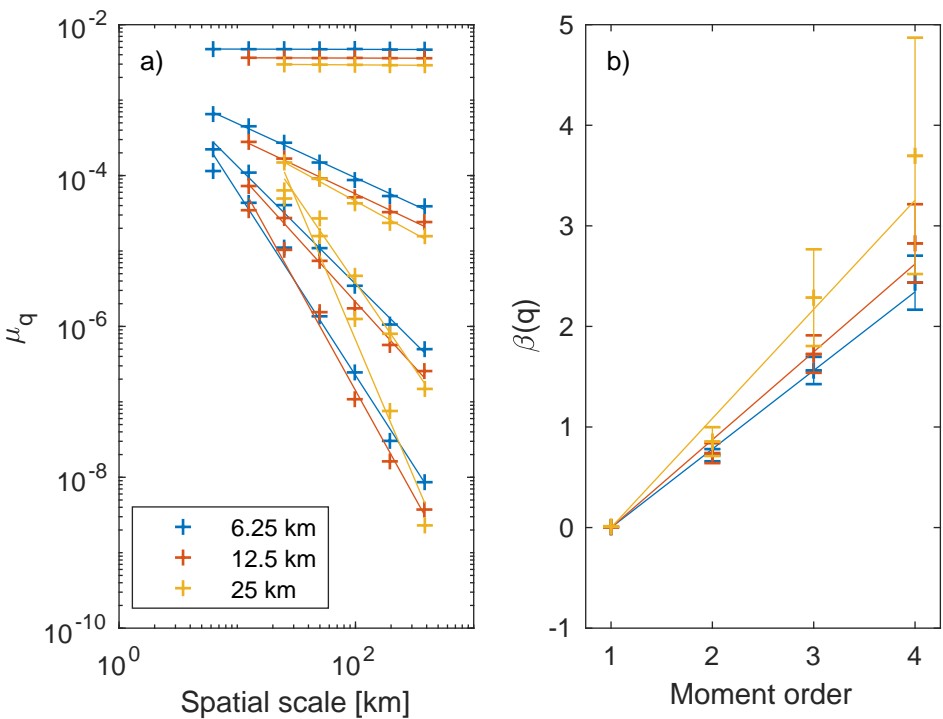

**Figure 7.** The spatial scaling and structure function of modelled lead fraction in the "Central Arctic" region over JFM, 2007. The two panes are the same as Fig. 3, only here the colours denote results from runs at different model resolutions.

is also the case at 12.5 km model resolution, even though the uncertainty is higher and the structure function is smaller. Going
from the 12.5 km to the 25 km model resolution continues the shift in the structure function going from 6.25 km to 12.5 km, of decreasing slopes and increasing uncertainty. Consequently, there is low confidence that the structure function indicates a multifractal scaling at the 25 km resolution. This change mirrors the change in fractality for the lead fraction going to 25 km from 12.5 km resolution.

## 6   Discussion

We have shown that the observed and modelled lead-fraction statistics in the Arctic have a monofractal character, as revealed by a spatial scaling analysis. The fact that the lead-fraction statistics are fractal means that the event propagation mechanism that generates fractal deformation rates also generates fractal lead fraction - which is to be expected. We note that the lead fraction has a monofractal character, while the deformation rates are multifractal. This means that while large deformations are more localised than small ones, all leads are localised equally. We did not investigate the difference especially, but a likely
explanation is that the lead fraction has too little variation to give multifractal statistics - it is essentially in either an on or off

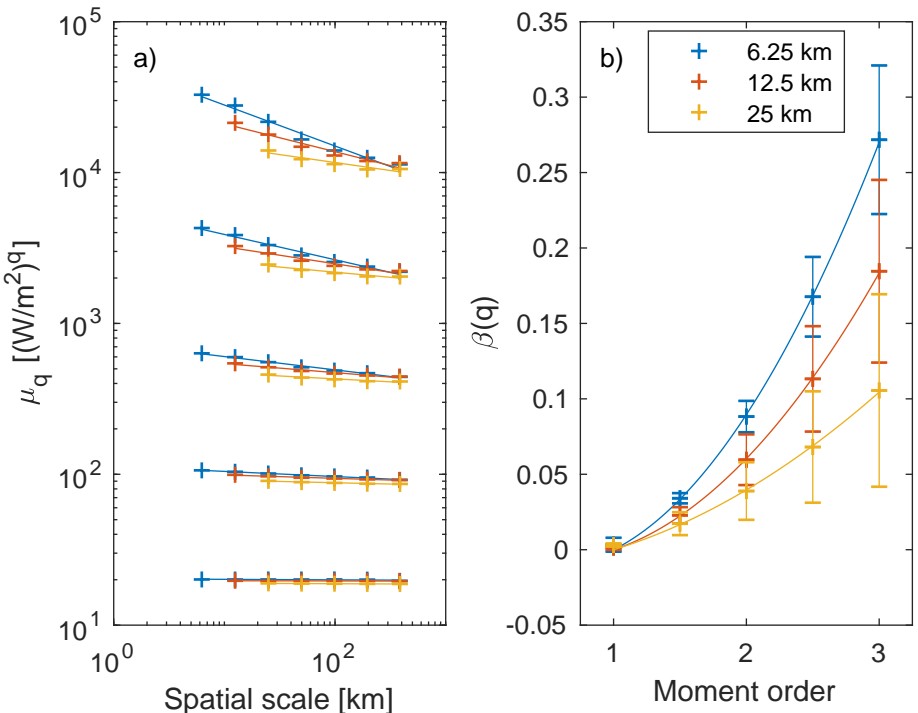

**Figure 8.** The spatial scaling of modelled heat fluxes in the "Central Arctic" region over JFM, 2007. The two panes are the same as Fig. 5, only here the colours denote results from runs at different model resolutions.

state. The modelled heat fluxes, on the other hand, do show a multifractal scaling, and this is probably because the heat flux is much more sensitive to the thickness of newly formed ice then the lead fraction.

We compare the fraction of pixel area covered with open water or thin ice within a lead as estimated from passive microwave satellite data to the modelled fraction of open water and newly formed ice thinner than 0.098 m in neXtSIM. Our analysis
shows that the agreement between the two is very good, and improves when we take into account the fact that small leads are under-represented in the observations. This under-representation of small leads is most likely the largest source of uncertainty in the present comparison. It is not clear whether the model needs further tuning or enhancement at this point.

The next step would be to compare neXtSIM with other lead fraction datasets at higher resolution, e.g., from Synthetic Aperture Radar (SAR) images or the Moderate Resolution Imaging Spectroradiometer (MODIS, see Willmes and Heinemann
(2015)). Such comparison is however considered to be outside the scope of the present paper. Our conclusion from our model evaluation is that the statistics of simulated lead fraction are well enough represented in neXtSIM to allow analysing the effects of lead opening on the heat fluxes using this model.

Considering the PDF of simulated heat fluxes, we note that we obtain a linear tail on a log-log scale in the Central Arctic region and a linear tail on a semi-log scale in the Arctic region. Using the results of the viscous model simulations, we show

that the log-log tail in the Central Arctic disappears when no leads are present, while much of the semi-log tail in the Arctic region remains. This suggests that the log-log tail in the Central Arctic and the observed scaling of heat fluxes is directly related to the formation of leads there. The semi-log tail in the Arctic can then be related to polynya formation, which does occur in the viscous model.

To explain how polynya formation causes the semi-log tail in the Arctic we use a simple model of ice growth in a polynya. This model assumes the polynya to be instantaneously opened by the wind and then closed due to ice growth where all ice that forms inside the polynya is herded by the wind to its down-wind edge. We can then assume that the closing rate is directly proportional to the area of the polynya since the total heat loss and total ice formation can be assumed to be proportional to the area of open water. An equation for the area evolution of the polynya is then

$$-\frac{dA}{dt} \propto A, \tag{2}$$

where $t$ denotes time and $A$ the polynya area. The solution to this ordinary differential equation is

$$A = A_0 e^{-tC}, \tag{3}$$

where $A_0$ is the initial polynya area, and $C$ is the growth rate of ice in the polynya. Letting $A$ evolve over time, we can see that the resulting PDF, $P$, has an exponentially decaying tail. Since the PDF is the normalised derivative of the cumulative distribution function, which is, in turn, an integral of $A$ from 0 to $\infty$, we have

$$P(x) = Ce^{-xC}. \tag{4}$$

Since we can assume the heat flux is proportional to the polynya area, to first order, we can expect the PDF of the heat flux to follow the same basic behaviour.

The linear tail on a log-log plot observed in the Central Arctic in Fig. 4a does not extend to the very highest simulated heat flux, as we would maybe expect. The reason for this discrepancy was not identified, but two likely explanations can be readily put forth. On the one hand, this effect could be physical; the largest fluxes produce ice the quickest, and hence we could expect this rapid ice growth to quickly dampen the heat flux and prevent the linear relationship from appearing. On the other hand, this could be a model artefact; we can imagine the sub-grid scale heterogeneity of newly formed ice to play a larger role for large heat fluxes than for small ones, and we know this heterogeneity to be only crudely parameterised in the model. Determining which of these explanations is true, and understanding their physical impact, is non-trivial and will not be attempted here.

Having analysed the PDF of heat fluxes for both the Arctic and Central Arctic regions we focus on the Central Arctic, where we found a clear multifractal scaling of the simulated heat flux. It is interesting that the heat-flux scaling is multifractal while the lead-fraction scaling is monofractal. This difference shows that, in the case of heat fluxes, the high values are also more localised in space, as is the case for sea-ice deformation (Marsan et al., 2004; Rampal et al., 2008; Stern and Lindsay, 2009; Bouillon and Rampal, 2015b; Rampal et al., 2019). This behaviour is to be expected, to some extent, because the largest heat fluxes depend not only on the lead fraction but also on the fraction and thickness of the newly formed ice in the leads. There is, therefore, additional information that goes into the heat-flux calculation, compared to the lead-fraction calculation, giving the potential for a multifractal scaling.

The current model does not take lead width into account when calculating the heat flux, as suggested by Andreas and Cash (1999). If it did, we would expect even stronger multifractality, as the highest heat fluxes would be even more localised. With the current approach, however, the presence of multifractality indicates the importance of properly localising the heat fluxes, since it is at these locations that the largest heat fluxes also occur. The best platform to investigate these effects is, of course, a coupled ice–atmosphere model, where the localisation of heat fluxes can influence the atmosphere in a manner similar to what Esau (2007) and Marcq and Weiss (2012) have started exploring. Coupling the ice with an ocean model is likely to have smaller effect at the time scales we consider here. Over a single winter the slab ocean model should be a sufficient simulation of the heat reservoir delivered by the ocean surface mixed layer. At longer timescales though, a coupling with the ocean likely becomes necessary to represent the evolution of the mixed layer and halocline.

We also briefly explored the impact of model resolution on the simulated lead fraction and heat fluxes and found that the statistics of the simulated lead fraction and heat fluxes are affected by the model resolution. This can, at least partially, be traced to the fact that neXtSIM does not exactly preserve deformation-rate (and in particular the opening-rate) scaling for the second and third moments and the structure function across different model resolutions (see figure 12 of Rampal et al., 2019). We can assume that the model's ability to reproduce the observed lead-fraction is due to its ability to reproduce the observed opening rates. Thus, if the scaling of the modelled opening-rates are not preserved across different model resolutions then we should expect the resulting lead fraction and heat flux scaling also not to be preserved. However, considering that the model does conserve the scaling for the first moment (the mean) of the deformation rates (see figure 12a of Rampal et al., 2019)) across different resolutions but not that of the lead fraction and heat fluxes, it is also likely that a lack of sub-grid scale heterogeneity of the ice thickness distribution in the model plays an important part in its inability to preserve the lead-fraction and heat-flux scaling and structure functions across different model resolutions.

The sub-grid scale thickness distribution used in neXtSIM is a very crude approximation of the highly heterogeneous thickness distribution present in reality. Furthermore, running the model at a high resolution should capture heterogeneity of the ice that would otherwise be parameterised in a coarse resolution model. As such, even if the deformation scaling were preserved across different model resolutions, one would not necessarily expect the lead-fraction or heat-flux scaling to be preserved. Realistically, the only way to achieve this would be to use a much better parameterisation of the sub-grid scale thickness distribution.

The main implications of these model shortcomings can be expected to be related to the influence an underestimation of the higher order moments of the heat fluxes has on the atmosphere and ocean in a coupled setup. It means that while different model resolutions will deliver the same mean heat fluxes and the same heat-flux statistics on the largest scales, we can expect extreme events to be underestimated in a coarse resolution model. This in turn may lead to an underestimation of local effects, such as mixing, moisture transport, and brine release — the large scale impact of which cannot be estimated here.

## 7  Summary and conclusions

We have analysed the observed lead fraction in the Arctic using scaling analysis. We then evaluated the simulated lead fraction in neXtSIM against the observed lead fraction. Following this we investigated the scaling of heat fluxes in neXtSIM and the conservation of lead-fraction and heat-flux scaling across different model resolutions. The main results of this work are:

- The observed lead fraction statistics in the Arctic have a monofractal scaling in space.

- The model reproduces the PDF and scaling of observed lead fraction in the Central Arctic reasonably well, especially
after taking the limits of the observations into account to the extent possible. It is not clear to which extent the differences between simulated and observed lead fractions are due to model or observation deficiencies.

- The model shows a clear multifractal scaling in space of the simulated heat fluxes in the Central Arctic. This scaling was shown to originate in the formation of leads.

- The mean heat flux is preserved across different model resolutions.

- The simulated lead-fraction and heat-flux scaling is not preserved across different model resolutions. This is most likely due in part to the misrepresentation of sub-grid scale heterogeneity in the current model ice thickness distribution.

These results indicate that the multi-fractal fracturing mechanism, already identified in sea-ice deformation rates (Weiss and Marsan, 2004; Marsan and Weiss, 2010), plays a significant role in lead formation. The multi-fractal character of the simulated heat fluxes shows that, in the model high heat fluxes are more localised than lower ones. One would expect higher peak heat
flux from leads that expand rapidly, so this behaviour is consistent with the fact that high deformation rates are more localised than low ones (Weiss and Marsan, 2004, e.g.,). The fact that the model reproduces well the observed lead fraction statistics gives us some confidence in the resulting simulated heat fluxes, but it is to be borne in mind that a fully coupled atmosphere–ocean–ice model would give a more complete picture of the coupled processes. Indeed, the role of leads in Arctic weather and climate remains largely unaddressed, but improved representation of leads in sea-ice models is an important step towards
rectifying this. Finally, our results indicate the relevance of a sub-grid scale parameterisation of ice-cover heterogeneity when investigating the statistics of lead fraction and heat fluxes through leads.

*Code and data availability.*  The lead fraction data set was not made publicly available when the original paper was published. The authors will provide copies of these data on request. The neXtSIM source code is not yet publicly available, but the code will be made available upon request, pending usage restrictions.

*Author contributions.*  EO and PR developed the scientific questions and analysis approach. EO performed the model runs and analysis and wrote the bulk of the text with PR and VD giving substantial input on the interpretation of results and the final text.

*Competing interests.* The authors declare that they have no conflict of interest.

*Acknowledgements.* This work was supported by the FRASIL (RCN #263044) and Nansen Legacy (RCN #276730) projects, financed by the Norwegian Research Council and the AOI project, financed by the Bjerknes Center for Climate Research. The authors would like to thank Nils Hutter and an anonymous second reviewer for reviewing the paper and making helpful suggestions and comments, as well as the editor, Petra Heil and the editorial staff at The Cryosphere for their assistance.

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
