# Peer review of "On the statistical properties of sea ice lead fraction and heat fluxes in the Arctic"

_The Cryosphere, 2020_

## Referee Comment (RC1) · Anonymous Referee #1 · 25 Feb 2020

This manuscript compares the statistical properties of sea ice lead fraction simulated by the neXtSIM sea ice model to satellite observations, then presents heat fluxes simulated by the model, discusses the impact of model resolution on results, and proposes a physical explanation for the semi-log tail in the probability distribution of simulated heat fluxes. The manuscript is novel and generally well-written, although some statements require citations (noted below). I recommend this manuscript for publication if my major comments below are addressed.

The manuscript would benefit from some further discussion for the general reader that better situates the findings in the context of the published literature and suggests implications for the broader scientific community, rather than simply for the development of neXtSIM. For example, why is it the mono/multifractal scaling interesting? Does the

[Figure]

work suggest parametrizations that could be used in traditional climate/sea ice models? This comment is merely a suggestion and the authors are not required to address it.

Major comments

I find the exclusion of the Beaufort Sea from the model-observation comparison to be poorly motivated. The model-observation comparison does not seem fair if the authors exclude regions where the comparison is poor. The authors need to motivate the exclusion of this region better and quantify to what extent the exclusion of the region affects their conclusions, or refrain from excluding it in their analysis.

Data availability – According to The Cryosphere data policy, 'Authors are required to provide a statement on how their underlying research data can be accessed.' This is missing from the manuscript.

Line-by-line comments

L7 Wasn't the Central Arctic region chosen to avoid the presence of polynyas?

L18 'In particular . . .' – add citation(s)

L19 'Leads . . . are a much more temporally and spatially clustered gateways' – this is unclear, please revise

L25 'causes' .. 'causing' – repetitive, rephrase

L32 Missing parentheses

L33 What area did the satellite image cover?

L36 'accurately reproduce the properties of lead fraction statistics' - add citation(s)

L43 In what situations does the mixed layer deepen in response to brine rejection? My understanding is that this is what occurs in low resolution models. Barthélemy et al. (2015) (https://doi.org/10.1016/j.ocemod.2014.12.009) could be cited here.

L48 'is actively being researched' – add citation(s)

L49 'Lead formation is closely linked . . .' – add citation(s)

L54 Suggest defining multifractality here.

L55 'This fundamental property . . .' This sentence needs more explanation for the general reader.

L74 Please provide more details on the slab ocean. Does it include any representation of ocean currents? How might the simplicity of the modelling configuration affect results?

L95 Define 'node' and 'cohesion'

L99 'The deficiencies of the linear viscous model are well known' – add citation(s)

L109 How is it an improvement?

L126 Why are the heat flux magnitudes provided as snapshots rather than daily means?

L154 Reword 'for future works' to 'in future work'

L167 'that gives good statistics.' What does this mean?

L168 Why not simply use a threshold of 10 cm? How much does the choice of threshold affect the results?

L182 Why do you think the model does not capture this?

Fig. 1 caption – define the red dashed lines. 'read' -> 'red'

L191 'excellent agreement' The figure is in log space, so some of the model-observation differences seem not insubstantial.

Fig. 2 caption – are these lines excluding the Beaufort Sea? Typo in 'Arctic'

[Figure]

L210 What is 'proper' spatial scaling?

L275 'after some algebra' – this wording is too casual for a journal article

L315 Reword 'this model shortcomings' to 'these model shortcomings'

―――――――――――――――――――

---

## Referee Comment (RC2) · Nils Hutter (Referee) · 10 Mar 2020

Review of "On the statistical properties of sea ice lead fraction and heat fluxes in the Arctic" by Olason et al. by Nils Hutter

In the manuscript, the authors use the statistical properties, heavy-tails and spatial scaling, that have been previously used to study the localisation of sea-ice deformation to study the lead density simulated by the sea-ice model neXtSIM and observed from satellite. The authors find that both agree with respect to these statistics. Furthermore, the model is used to study the same statistical properties for simulated heat fluxes with the result that heat fluxes are strongly localised. This analysis contributes to the on-going research how important small-scale information of sea-ice (floes, leads and

pressure ridges) are for the interaction of sea-ice with the atmosphere and ocean. The paper is written concisely and I recommend it for publication if the following comments are addressed.

General comment:

The authors choose rather complex statistical tools by analysing the heavy-tails of PDFs and the spatial scaling. These methods are appropriate to study the localisation of lead density and simulated heat flux, which is the main topic of the manuscript. However, a comparison of the spatial distribution of lead density as done in Wang et al. (2016) and Hutter & Losch (2020) is missing, although all data would be available for that. In Fig. 1 such a comparison is made for a snapshot of a single day. I recommend to add a comparison of spatial distribution of lead-density for the entire winter analysed in this paper (maybe replacing Fig. 1). In doing so, the model evaluation of this manuscript would be more comprehensive by showing that the model (might be)/is able to reproduce the large-scale spatial distribution and the strong localisation of lead-density.

Specific comments:

P2, line 32: "Andreas and Cash (1999); Esau (2007)" - wrong citation style

P2, line 35: "including smaller leads increased by 55% the total estimated heat flux" - including smaller leads increased the total estimated heat flux by 55%.

P2, line 35-37: I assume that the magnitude of the overall heat flux is adjusted by the tuning of thermodynamic parameters in coarse resolution climate models. However the spatial distribution and local magnitude might be off, if leads are not resolved in these models. Please clarify.

P2, line 48-49: "the statistical properties of leads in large-scale sea-ice models have not yet been shown to be robustly reproduced" - How about Wang et al. (2016) and Hutter & Losch (2020). Wang et al. (2016) shows agreement in the lead density in the Arctic

between a model simulations and satellite observations. Hutter & Losch (2020) show that multiple spatial and temporal properties of LKFs, which are leads and pressure ridges, observed from satellite are matched by large-scale sea-ice simulations.

P2, line 61: "Section 2.1" - Section 2.1 presents only the model set-up. Please refer to Section 2.

P3, line 61-70: This paragraph reads a bit wordy. Maybe consider to rephrase it.

P3, line 83: "model mesh" - Model mesh or the mesh to which the model output is interpolated?

P4, line 1-2: Not clear, from which data product concentration and from which product thickness is taken. Please clarify.

P5, l 125 "order" - order -> orders

P5, l 136: "2011" - 2011 vs model year 2007? In the model description it is written that the model is ran for winter 2007, later on in the paper you evaluate only the year 2011. Please clarify. Does this sentence anyways not rather belong to the results section?

P5, l150: "PÌĎ âĹij L$-\beta$(0)" - Supposing x_bar should represent the mean, it should be beta(q=1). For q=0 no scaling should be observable, if equation (2) is used (x^0=1 for all samples).

P6, l152: "Stern et al. (2018) argue that this method provides a reasonably accurate estimate of the power-law fit." - In addition, Stern et al. (2018) argue that no matter what method is used for estimate of the power-law exponents a goodness-of-the-fit test like in Clauset et al. (2009) should be performed. Please clarify, if you do such a test, or why it is not necessary in this case.

P6, l153: "might provide" - Replace by "provides". Both Stern et al. (2018) and Clauset et al. (2009) say it provides better estimates. Given that the method is computationally not much more expensive, it is unclear to me, why you choose to use a more inaccurate

method.

P6, l171-172: "It is important to note that the simulated lead fraction is not strictly a lead fraction as it includes all open water areas, including polynyas (cf figure 1)." - How about using a smoothened concentration field to mask large open-water areas as around Svalbard?

P6, L 177: "showing a deviation from linearity at around 70%" - I can not see a clear deviation. Is it due to the dashed line style. An annotation to the plot could help to point the reader to what you mean.

P6, l178: "When excluding this region, the observations also show a linear decrease (Fig. 2, solid blue line)." - This does not fit to the caption of the Figure2 (dashed for "Arctic" and solid for "Central Arctic").

P6, l 182: "However, we suspect that the large number of small leads forming there may result in increased noise in the lead fraction product (see Fig.1) and an overestimation of the large lead fractions." - Not clear to me. Please be more specific why more small leads lead to an overestimation of the lead fraction product. Or do you mean that by having many small leads the lead fraction increases, but the model does not resolve these small leads and therefore shows lower lead fractions?

P8, l197: "than 20%" - Add (note shown) or reference to figure.

P8, l 203: "strong indicator" - Be cautious, even if the scaling is right, the regional distributions could be off, i.e. high lead fractions found close to the coast or in Beaufort sea in observations could be reproduced by the model in different regions. To clarify this, please be more specific what you mean with lead-fraction patterns in the text.

P11, l 243: "In addition to these differences in the scaling, there also seems to be a difference in the nature of the structure function, depending on the model resolution" - Please also discuss the change in structure function for the lead fraction. It appears that the linear fit is not appropriate to fit the structure function of the coarse resolution

models (The fit does not pass the uncertainty interval for q=1).

P13, l267-269: "We also assume that the closing is directly proportional to the area of the polynya since most of the heat loss and ice formation happens over open water." - This assumption is not clear: I agree that ice formation is larger over open water, but if a polynya is formed instantaneously the entire area of the polynya starts to freeze at the same time. Please clarify.

P14, l 219: "figure 4" - Please reference the subfigure for clarity.

P15, l300-308: "This is partially due to the fact that neXtSIM . . . the lead-fraction and heat-flux scaling and structure functions across different model resolutions." - This paragraph is not clear to me. It is difficult to follow your line of argumentation. Please revise and rephrase.

P16, l320: "Conclusions" - You provide rather a summary of the paper than a conclusion. So, please change the title of the section accordingly.

Data and code availability: A statement is missing, where to find the code and data of this study.

Figure 1: "Lead fraction larger than 0.05 is indicated in yellow." - Why do you show the thresholded fields instead of using a colormap that highlights the 0.05 fraction about shows the entire range of lead fractions? I recommend to use a show the entire range of lead fractions.

Figure 2: "The dashed straight lines are linear fits discussed in the text." - Could you use color to indicate which fit belongs to which data. Please use different linestyle for the fits and the "Arctic". Please also add all lines to the legend to clarify. In the caption "Arcitc" should be "Arctic".

Figure 4. Please add (a) and (b) labelling to the subfigures.

Figure 7: "he" to "the". How do you choose the order of the polynomial fit of the

structure functions here? For 12.5km and 25km the linear fit does not seem to be appropriate to fit the structure function, but rather a higher order (quadratic?) is required. This, however, would mean that the lead fraction gets multifractal for higher model resolution. Please elaborate on this difference when changing the model resolution in the text.

References:

Wang, Q., Danilov, S., Jung, T., Kaleschke, L., and Wernecke, A.: Sea ice leads in the Arctic Ocean: Model assessment, interannual variability and trends, Geophys. Res. Lett., 43, 7019–7027, https://doi.org/10.1002/2016GL068696, 2016.

Hutter, N. and Losch, M.: Feature-based comparison of sea ice deformation in lead-permitting sea ice simulations, The Cryosphere, 14, 93–113, https://doi.org/10.5194/tc-14-93-2020, 2020.

---

## Author Comment (AC1) · 16 Jun 2020

The manuscript would benefit from some further discussion for the general reader that better situates the findings in the context of the published literature and suggests implications for the broader scientific community, rather than simply for the development of neXtSIM. For example, why is it the mono/multifractal scaling interesting? Does the work suggest parametrizations that could be used in traditional climate/sea ice models? This comment is merely a suggestion and the authors are not required to address it.

This is a very insightful comment as it goes to the heart of the motivation of the paper. Both this comment and comments by the other reviewer show that the paper's motivation needs to be stated more clearly - something we have tried to do in the modified manuscript.

Major comments
I find the exclusion of the Beaufort Sea from the model-observation comparison to be poorly motivated. The model-observation comparison does not seem fair if the authors exclude regions where the comparison is poor. The authors need to motivate the exclusion of this region better and quantify to what extent the exclusion of the region affects their conclusions, or refrain from excluding it in their analysis.

Indeed, the motivation for excluding the Beaufort Sea was not clear in the original submission. Our main reason for doing so was an apparent inconsistency in the observations between the Beaufort Sea and the rest of the Central Arctic. Further work on this issue - following this review - has, however, made it clear to us that excluding the Beaufort Sea is not well justified. In the revised text all analysis and comparisons are made on the "Arctic" and "Central Arctic" regions only.

Data availability – According to The Cryosphere data policy, 'Authors are required to provide a statement on how their underlying research data can be accessed.' This is missing from the manuscript.

We've added this to the revised manuscript.

Line-by-line comments
L7 Wasn't the Central Arctic region chosen to avoid the presence of polynyas?
        It should have said wider Arctic - this has been corrected in the revised text
L18 'In particular . . .' – add citation(s)
        Done
L19 'Leads . . . are a much more temporally and spatially clustered gateways' – this is unclear, please revise
        We've rewritten the sentence as "Leads in the centre of the Arctic Basin, on the other hand, are much more difficult to study because they are much narrower and shorter-lived than polynyas, and at the same time can form anywhere in the Arctic Basin."
L25 'causes' .. 'causing' – repetitive, rephrase
        Changed 'causes' to 'drives'
L32 Missing parentheses
        Fixed
L33 What area did the satellite image cover?
        It was a 60x66 km2 image, we've added this to the text.
L36 'accurately reproduce the properties of lead fraction statistics' - add citation(s)

We've rewritten this statement to be more accurate and added references

L43 In what situations does the mixed layer deepen in response to brine rejection? My understanding is that this is what occurs in low resolution models. Barthélemy et al. (2015) (https://doi.org/10.1016/j.ocemod.2014.12.009) could be cited here.

The mixed layer shoals when (relative) ice velocity is low and the brine forms a plume that sinks to the bottom of the mixed layer. When the ice velocity is high the shear induces mixing of the brine which in turns causes a deepening of the mixed layer. In (most) low-resolution ocean models the brine is released uniformly into the first ocean layer causing mixing and deepening of the mixed layer - similar to the high-ice-velocity regime. Nguyen et al showed that releasing the brine at the bottom of the mixed layer improves the simulation. This is described in the text, although we have modified it slightly to make it clearer.

We were not aware of the Barthélemy paper, but it was very interesting to see the importance of including both the low- and high-velocity regimes. So we've added a reference to that as well. Thank you for pointing it out.

L48 'is actively being researched' – add citation(s)

Done

L49 'Lead formation is closely linked . . .' – add citation(s)

This sentence was poorly formulated. We've reformulated it to read "When sea ice deforms ridges and leads are formed." - which doesn't require a citation.

L54 Suggest defining multifractality here.

We have added a discussion of the relevance of (multi)fractality here.

L55 'This fundamental property . . .' This sentence needs more explanation for the general reader.

We've removed this sentence in favour of the more detailed discussion added.

L74 Please provide more details on the slab ocean. Does it include any representation of ocean currents? How might the simplicity of the modelling configuration affect results?

We intentionally didn't go into much detail about the model - including the slab ocean - since this is presented in more detail in Rampal et al (2019) and Rampal et al (2016). We did still add the sentence "Oceanic heat loss results in lowering of the slab ocean temperature, which may be compensated for by new-ice formation and nudging of the slab ocean layer temperature to reanalysis results." to the text and hope you find it sufficient to address your comment.

As for how using a slab ocean affects the results then we expect this to be minor. We expect a larger contribution from the atmosphere in this respect, and that is mentioned in the discussion. To address your comment we've added a few sentences to this effect in the discussion section.

L95 Define 'node' and 'cohesion'

'Nodal spacing' was left over from a previous version of the manuscript. The correct phrasing is simply '12.5 and 25 km resolution', and we have changed this in the revised manuscript. Mentioning the cohesion at this point borders on being too precise, in our opinion and we prefer to refer to the discussion in Bouillon and Rampal w.r.t. scaling of cohesion. We have slightly reformulated and rephrased to make this clearer.

L99 'The deficiencies of the linear viscous model are well known' – add citation(s)

Done

L109 How is it an improvement?

> The new product fixes an overestimation of the lead fraction present in the original. We've added a sentence to that effect in the text.

L126 Why are the heat flux magnitudes provided as snapshots rather than daily means?

> They are not - this was left over from an earlier version of the paper. The heat fluxes and lead fractions are both daily means.

L154 Reword 'for future works' to 'in future work'

> Done

L167 'that gives good statistics.' What does this mean?

> It just means that it gives the same slope of the PDF, as per the latter part of the sentence. It now reads "We, therefore, choose a threshold thickness for the model that has the same slope of the PDF as the observed one, as shown below."

L168 Why not simply use a threshold of 10 cm? How much does the choice of threshold affect the results?

> We wanted to see if we could deduce the appropriate threshold from the model statistics. The fact that the two are very close shows that the processes in the model are a reasonable reproduction of those in reality. Having done this it doesn't matter much which value we use for the comparison, the results are essentially the same. The paper also says that variations of the threshold of about 1 cm give the same results.
> Note, however, that the 10 cm threshold is more of an educated guess and probably not a fixed threshold anyway - there is also a dependence on emissivity, frost flowers, and probably other factors as well. Since the 10 cm threshold is a very rough estimate we see no reason to prefer that over the model deduced threshold of 9.1 cm.

L182 Why do you think the model does not capture this?

> We don't exclude the Beaufort Sea anymore, so this comment is not relevant.

Fig. 1 caption – define the red dashed lines. 'read' -> 'red'

> The dashed line was what we exclude as the Beaufort Sea, but this is not relevant anymore.

L191 'excellent agreement' The figure is in log space, so some of the model- observation differences seem not insubstantial.

> Indeed this was maybe overly enthusiastic wording. We go into more detail below about the model-observation differences to show that the observation shortcomings are such that it's hard to say to what extent the differences are indeed substantial. We, therefore, replaced 'excellent' with 'good' and added a sentence highlighting that the agreement between model and reality is probably much better than the comparison in the figure seems to indicate at a first glance.

Fig. 2 caption – are these lines excluding the Beaufort Sea? Typo in 'Arctic'

> Yes, but as we don't exclude the Beaufort Sea anymore, so this comment is not relevant.

L210 What is 'proper' spatial scaling?

> We've removed the word 'proper' - it should not have been there.

L275 'after some algebra' – this wording is too casual for a journal article

> We've removed the phrase 'after some algebra'

L315 Reword 'this model shortcomings' to 'these model shortcomings'

> Done

---

## Author Comment (AC2) · 16 Jun 2020

General comment:
The authors choose rather complex statistical tools by analysing the heavy-tails of PDFs and the spatial scaling. These methods are appropriate to study the localisation of lead density and simulated heat flux, which is the main topic of the manuscript. However, a comparison of the spatial distribution of lead density as done in Wang et al. (2016) and Hutter & Losch (2020) is missing, although all data would be available for that. In Fig. 1 such a comparison is made for a snapshot of a single day. I recommend to add a comparison of spatial distribution of lead-density for the entire winter analysed in this paper (maybe replacing Fig. 1). In doing so, the model evaluation of this manuscript would be more comprehensive by showing that the model (might be)/is able to reproduce the large-scale spatial distribution and the strong localisation of lead-density.

Indeed, we have not made the kind of spatial distribution analysis that Wang et al and Hutter and Losch did. This would indeed be an interesting additional metric for a model-observations comparison, but we feel that adding it to the present paper would not be appropriate. The reason being that in this paper we are introducing the notion of using spatial scaling analysis to investigate lead fraction patterns. This is motivated by the spatial scaling we see in deformation rates, as we know that lead formation is closely linked to deformation.

In this paper, we show that both observed and modelled lead fraction patterns demonstrate spatial scaling and we then use the model to show that the (modelled but unobserved) heat-flux patterns also demonstrate spatial scaling. The model-observations comparison is therefore only there so that we can with some confidence say that there is likely also a spatial scaling of the heat fluxes in reality.

Adding the spatial distribution analysis, as suggested, would therefore not add to the central theme of the paper - which should be the spatial scaling of lead fraction and heat fluxes. We feel that adding an arguably parallel discussion to the paper in this manner would not be beneficial for the paper, but make it less focused and more difficult to follow. We therefore respectfully decline to follow the reviewer's recommendation on this point. We do, however, feel that this comment, together with one of the other reviewer's, makes it clear that the introduction and motivation of the paper needs to be improved - something that we have attempted to do in the revised manuscript.

Specific comments:
P2, line 32: "Andreas and Cash (1999); Esau (2007)" - wrong citation style
        Fixed
P2, line 35: "including smaller leads increased by 55% the total estimated heat flux" - including smaller leads increased the total estimated heat flux by 55%.
        Fixed
P2, line 35-37: I assume that the magnitude of the overall heat flux is adjusted by the tuning of thermodynamic parameters in coarse resolution climate models. However the spatial distribution and local magnitude might be off, if leads are not resolved in these models. Please clarify.
        You are right. We have changed under-represented to misrepresented, which is more appropriate.

P2, line 48-49: "the statistical properties of leads in large-scale sea-ice models have not yet been shown to be robustly reproduced" - How about Wang et al. (2016) and Hutter & Losch (2020). Wang et al. (2016) shows agreement in the lead density in the Arctic between a model simulations and satellite observations. Hutter & Losch (2020) show that multiple spatial and temporal properties of LKFs, which are leads and pressure ridges, observed from satellite are matched by large-scale sea-ice simulations.

> We have now included those references in the text. The point we wanted to make was that models that are normally used to study the Arctic have neither the resolution nor the numerics necessary to resolve these features. This is made clearer in the revised manuscript.

P2, line 61: "Section 2.1" - Section 2.1 presents only the model set-up. Please refer to Section 2.

> Fixed

P3, line 61-70: This paragraph reads a bit wordy. Maybe consider to rephrase it.

> We have rewritten parts of this and hope that it reads better now.

P3, line 83: "model mesh" - Model mesh or the mesh to which the model output is interpolated?

> The model mesh. We have rewritten this sentence slightly to make this clearer.

P4, line 1-2: Not clear, from which data product concentration and from which product thickness is taken. Please clarify.

> This is clarified in the revised text

P5, l 125 "order" - order -> orders

> Fixed

P5, l 136: "2011" - 2011 vs model year 2007? In the model description it is written that the model is ran for winter 2007, later on in the paper you evaluate only the year 2011. Please clarify. Does this sentence anyways not rather belong to the results section?

> It was supposed to say 2007 and we've fixed that (here and in several other places). We used 2011 in a previous version of the paper. We think that the sentence belongs here because it illustrates the method of using averages of JFM and does not address the results. The reference to figures 2 and 4 is for illustrative purposes and the figures are only discussed in the results section.

P5, l150: "PÌD âLij L−β(0)" - Supposing x_bar should represent the mean, it should be beta(q=1). For q=0 no scaling should be observable, if equation (2) is used (xˆ0=1 for all samples).

> Yes, \bar{x} is the mean and it should be \beta(1), not \beta(0). We've fixed this.

P6, l152: "Stern et al. (2018) argue that this method provides a reasonably accurate estimate of the power-law fit." - In addition, Stern et al. (2018) argue that no matter what method is used for estimate of the power-law exponents a goodness-of-the-fit test like in Clauset et al. (2009) should be performed. Please clarify, if you do such a test, or why it is not necessary in this case.

P6, l153: "might provide" - Replace by "provides". Both Stern et al. (2018) and Clauset et al. (2009) say it provides better estimates. Given that the method is computationally not much more expensive, it is unclear to me, why you choose to use a more inaccurate method.

> For the two points above: The second half of this paragraph was wrong. The MLE method of Clauset et al is to estimate if the PDF follows a power law, but this is not our concern here. We know that the PDF has a "fat" tail (doesn't need to be a power-law) and then we can reasonably expect there to be a scaling relationship - which we

> do find. We've removed the erroneous portion of the paragraph and slightly rewritten the surrounding text without changing its contents or meaning.

P6, l171-172: "It is important to note that the simulated lead fraction is not strictly a lead fraction as it includes all open water areas, including polynyas (cf figure 1)." - How about using a smoothened concentration field to mask large open-water areas as around Svalbard?

> If we're doing that we would essentially be fabricating data since the smoothed field would have very different characteristics from the rest of the field.

P6, L 177: "showing a deviation from linearity at around 70%" - I can not see a clear deviation. Is it due to the dashed line style. An annotation to the plot could help to point the reader to what you mean.

> This was not very clear in the original submission. But as we don't exclude the Beaufort Sea anymore, so this comment is no longer relevant.

P6, l178: "When excluding this region, the observations also show a linear decrease (Fig. 2, solid blue line)." - This does not fit to the caption of the Figure2 (dashed for "Arctic" and solid for "Central Arctic").

> We've fixed the caption, which was wrong.

P6, l 182: "However, we suspect that the large number of small leads forming there may result in increased noise in the lead fraction product (see Fig.1) and an overestimation of the large lead fractions." - Not clear to me. Please be more specific why more small leads lead to an overestimation of the lead fraction product. Or do you mean that by having many small leads the lead fraction increases, but the model does not resolve these small leads and therefore shows lower lead fractions?

> Indeed, the motivation for excluding the Beaufort Sea was not clear in the original submission. Our main reason for doing so was an apparent inconsistency in the observations between the Beaufort Sea and the rest of the Central Arctic. Further work on this issue, following this review, has, however, made it clear to us that excluding the Beaufort Sea is not well justified. In the revised text all analysis and comparisons are made on the "Arctic" and "Central Arctic" regions only.

P8, l197: "than 20%" - Add (note shown) or reference to figure.

> This refers to the discussion in the paragraph above, so we've added "(see above)" to the text.

P8, l 203: "strong indicator" - Be cautious, even if the scaling is right, the regional distributions could be off, i.e. high lead fractions found close to the coast or in Beaufort sea in observations could be reproduced by the model in different regions. To clarify this, please be more specific what you mean with lead-fraction patterns in the text.

> It is right that our use of the words "lead-fraction patterns" is too broad and not supported by our results. We have therefore rephrased the sentence to read "... strong indicator that the model is simulating lead formation in a physical and realistic manner …"

P11, l 243: "In addition to these differences in the scaling, there also seems to be a difference in the nature of the structure function, depending on the model resolution" - Please also discuss the change in structure function for the lead fraction. It appears that the linear fit is not appropriate to fit the structure function of the coarse resolution models (The fit does not pass the uncertainty interval for q=1).

> This is true and we have noted it now.

P13, l267-269: "We also assume that the closing is directly proportional to the area of the polynya since most of the heat loss and ice formation happens over open water." - This

assumption is not clear: I agree that ice formation is larger over open water, but if a polynya is formed instantaneously the entire area of the polynya starts to freeze at the same time. Please clarify.

> This was indeed not clear enough in the initial submission. The point is that we can assume that the closing rate is directly proportional to the area of the polynya since the total heat loss and total ice formation can be assumed to be proportional to the area of open water. This leads to the differential equation and the rest of this simplistic model. We've modified the text to be more precise.

P14, l 219: "figure 4" - Please reference the subfigure for clarity.

> Done

P15, l300-308: "This is partially due to the fact that neXtSIM . . . the lead-fraction and heat-flux scaling and structure functions across different model resolutions." - This paragraph is not clear to me. It is difficult to follow your line of argumentation. Please revise and rephrase.

> The idea here is that if the opening-rate scalings are not resolution independent then we should expect the lead fraction (and heat flux) scalings also not to be resolution independent. We've modified the text to try to make this clearer.

P16, l320: "Conclusions" - You provide rather a summary of the paper than a conclusion. So, please change the title of the section accordingly.

> We've changed this to "Summary and conclusions"

Data and code availability: A statement is missing, where to find the code and data of this study.

> Done

Figure 1: "Lead fraction larger than 0.05 is indicated in yellow." - Why do you show the thresholded fields instead of using a colormap that highlights the 0.05 fraction about shows the entire range of lead fractions? I recommend to use a show the entire range of lead fractions.

> We now use a colormap that highlights the important range of lead fractions.

Figure 2: "The dashed straight lines are linear fits discussed in the text." - Could you use color to indicate which fit belongs to which data. Please use different linestyle for the fits and the "Arctic". Please also add all lines to the legend to clarify. In the caption "Arcitc" should be "Arctic".

> We've reworked this figure after not excluding the Beaufort Sea anymore, and in this case there's no need to differentiate between the two fits. We have added the all the lines to the legend.

Figure 4. Please add (a) and (b) labelling to the subfigures.

> Done

Figure 7: "he" to "the". How do you choose the order of the polynomial fit of the structure functions here? For 12.5km and 25km the linear fit does not seem to be appropriate to fit the structure function, but rather a higher order (quadratic?) is required. This, however, would mean that the lead fraction gets multifractal for higher model resolution. Please elaborate on this difference when changing the model resolution in the text.

> We've fixed the typo. We chose a linear fit for the structure function, same as in figure 3, and as you surmise. We chose this to be consistent with the analysis of the 6.5 km resolution run. Indeed the 25 km fit could be quadratic, but the uncertainty associated with the scaling at this resolution is so high that the significance of this is limited. This is now noted in the text.

References:

Wang, Q., Danilov, S., Jung, T., Kaleschke, L., and Wernecke, A.: Sea ice leads in the Arctic Ocean: Model assessment, interannual variability and trends, Geophys. Res. Lett., 43, 7019–7027, https://doi.org/10.1002/2016GL068696, 2016.

Hutter, N. and Losch, M.: Feature-based comparison of sea ice deformation in lead-permitting sea ice simulations, The Cryosphere, 14, 93–113, https://doi.org/10.5194/tc- 14-93-2020, 2020.

---

## Author Response (AR2)

Dear editor,

Thank you very much for your constructive comments. We have addressed all of them and we hope that our modifications are to your satisfaction.

Below we have copied out your comments in black and inserted our response in blue type.

Regards,

Einar (on behalf of the authors)

Comments to the Author:
I thank the author for their rewrite of this manuscript which resulted in a much approved version. Pls be guided by the new comments of referee #1 as well as those shown here to tighten up the manuscript in anticipation for publication.

Non-public comments to the Author:
General:
* I abbreviate "manuscript" with "ms".
* Throughout the ms, pls correct "e.g." to "e.g.,".

Done

* Apply consistent calling format: i.e., l 177 "(figure 2, dashed blue line)." versus l 178 "(Fig. 2, solid blue line)". Suggest the latter, final advise to come from TC technical editors team.

Done

Specific comments:
1: Suggest to avoid starting the abstract of a ms with "In this paper".

Fixed

17: Correct "e.g." to "e.g.."
20: Correct "e.g." to "e.g.." and throughout the remainder of this ms.

Fixed

25: As your ms is concerned with the sea-ice cover on the vast Arctic Ocean,
I am surprised you refer to icing up issues of man-made equipment, i.e.,
"such as aircraft, power lines, and roads". I suggest to remove this text.

Done

29: There is too much detail without relevance presented: "four cases of
capping inversion". Without going to the provided reference the reader
cannot know about the different inversion capping scenarios of the
referenced study. Rewrite "found that this kind of entrainment occurred

in all four cases of capping inversion they studied," with "identified
such entrainment for all scenarios they encountered,".

Done

34: Rewrite in better English: "such that including smaller leads increased
by 55% the total estimated heat flux."

Done

35: Remove "result".

Done

44: Replace "faithful" in "faithful simulation" with an appropriate descriptor

Replaced with "realistic"

48: "statistical properties of leads in large-scale sea-ice models have not yet
been shown to be robustly reproduce": Suggest to add a note on discrepancy
in spatial footprints... of grid resolution in "large-scale sea-ice model"
versus effective width of leads.

Changed to "… not yet been shown to robustly reproduce the statistical properties of lead
fraction – as large-scale models cannot simulate single leads, only the frequency of their
occurrence within a grid cell."

83: Correct "the Barents and Kara Seas" to "the Barents and Kara seas".

Done

89: Change "We start the model on November 15th," to We start the model run on
November 15th,"

Done

90: "Arctic Ocean is fully ice-covered": Technically an ocean basin never fully
ice covered. For that all "grid cells" would require 100% ice concentration.
91: Need to clarify what is meant by "uniform" in "hence the mechanical and
thermodynamical regimes are approximately uniform."

We have rewritten this sentence to read "… 2007, so as not to influence the results by the
very different heat fluxes and lead fractions seen during the freeze-up and melt periods",
thus addressing also the comment above.

114: Correct "The data shows" to "The data show" as data is plural.

Done

117: Pls use SI units. For example to correct "10 cm as thin ice" to "10 cm as thin ice"

We expect you mean for us to change to "0.1 m as thin ice". However, cm is an SI unit and using cm in this case infers a sense of scale; using cm indicates that the physical properties considered are on the range of centimetres, not metres or kilometres. Indeed, the resolution of the model and observations is quoted in kilometres, not metres, for this very reason. We have, therefore, not changed this.

123: Correct "(i.e." to "(i.e.,".

Done

123: Change "there is potentially a fractal spatial scaling to be found for the quantity." to "then there is potential that the investigated variable is subject to fractal scaling."

Done

126: Correct "i.e. all" to "i.e., all".

Done

136: Is this a word? "coarse-graining:>Lw'

Yes, see e.g., Bouillon and Rampal (2015a), Rampal et al. (2016,2019), or Dansereau et al. (2016).

126; Change "at varying spatial scales, to "across a range of spatial scales,".

Done

147: Correct "we produce plots of the moment values" to "we derive the moment values".

Done

147 and an earlier instance: In section "2.3 Methodology" results should NOT be presented.

Fixed.

172: Correct "ata is filtered" to "data are filtered".

Fixed

181: Use of "destroys" appears inappropriate. Pls replace with a more suitable verb.

Replaced with "… and prevent the linear relationship from appearing."

197: Replace "Perhaps" with "We suggest".

Done

198: Correct "However it would" to "However this would".

Done

208: Change "30 W/m 2 and 110 W/m 2 ," to "30 and 110 W/m 2 ,".

Done

220: Add info to read"the tails of the distribution of xxx".

Done – here it refers to the distribution of heat fluxes

228: Remove sentence "The results of the scaling analysis for all three model resolutions are shown in figure 7 for the lead fraction and figure 8 for the heat flux."

Done

229: Add citation for repective figures (in brackets).

It is not clear to us which citation should be added. The line in question reads ".. all three model resolutions are shown in figure 7 for the lead fraction and figure 8 for the heat flux. The comparison shows that …". Here the numbers 7 and 8 refer to figures in this paper.

231: Change "between different resolutions" to "for the chosen resolutions".

Done

231: Going from 20% to 30% is a one tenth change, hence not "small". --> Reconsider statement and rewrite.

We have removed the qualification from this sentence.

238: Correct "All three models" to "All three models realisations".

Done

239: Avoid "difference" and "different" in same sentence/short succession.:

We've removed the word "different" in "different model results" as it was redundant.

247: Change "As a result, we cannot say with any confidence that the structure function indicates a multifractal scaling at the 25 km resolution." to "Consequently, there is low confidence that the structure function indicates a multifractal scaling at the 25 km resolution."

Done

255: Correct ", e.g." to ", e.g.,".

Done

260: Write statements as "we obtained" in the present tense, i.e., "we obtain". Apply all through the ms.

Done

266-278: Strongly suggest to move the "Polynys results" into the "Results chapter", i.e., before the "Discussions".

We don't wish to move the paragraphs in question to the results section. In our opinion they are discussing and explaining the results, not merely describing them. This belongs in the discussion section, although in this case a small introductory paragraph is needed to orient the reader as to what it is we will now discuss.

Fig. 8, Caption: Correct "here he colour" to "here the colour".

Done

275: This writing style is too casual "we have after some algebra". Please rewrite.

Done

324: The statement "very well" is not supported by the results and discussion shown earlier.

This has been rephrased in accordance with a request from reviewer #1.

335: Suggest to acknowledge here "external data" used in this study.

Done

335: Suggest to also acknowledge the review and comments furnished by the two reviewers.

Done

Dear reviewer,

Thank you very much for a constructive review. We have addressed all your comments and we believe this has improved the paper further. We hope that you agree and that our modifications are to your satisfaction.

Below we have copied out your comments in black and inserted our response in blue type.

Regards,

      Einar (on behalf of the authors)

Main points to be addressed prior to publication:

Can you be sure that the lack of small leads in the observations explains all of the difference between model and observations up to 40% lead fraction? It is also possible that the model is biased at lead fractions below 40%. I don't think the statements at L230 that
- 'agreement between model and reality is probably better than a first-order interpretation of figure 3 would suggest'; and
- 'Perhaps a fairer comparison between the model simulation and observations would therefore consider only lead fractions larger than 40%'

are justified based on what has been presented in the manuscript.

It is correct that we can indeed not be sure that the lack of small leads in the observations explains all the difference between model and observations up to 40% lead fraction. The point we wanted to make was that the lack of small leads below c.a. 40% makes evaluation of model with this data difficult. We have tried to clarify this by replacing the sentence "This discrepancy appears because small values of the lead fraction are under-represented in the observations" with "We know that small leads are not detected in the satellite product and we should, therefore, only compare model and observations in the high lead-fraction range, where we know the observations to be reliable".

In a similar vein, there is no justification presented for claiming that there is 'noise' in the observations (Fig. 3 caption).

Yes, this statement requires more justification. We have simply removed it, since it is not important for the contents of the paper.

The authors conclude that, 'The model reproduces the PDF and scaling of observed lead fraction in the Central Arctic very well.' (L383) This seems too strong, and I think it should be rephrased to something like 'shows agreement with observations for lead fractions over 40%'

We agree that this is too strong of a statement. We have replaced it with "The model reproduces the PDF and scaling of observed lead fraction in the Central Arctic reasonably well, especially after taking the limits of the observations into account to the extent possible". We feel that explicitly mentioning the 40% limit is too detailed here, especially since that limit is only approximate.

The following points aren't necessary to address, but the authors may wish to consider:

Regarding my previous comment on the thin ice threshold-
It seems somehow artificial to me to compute the thin ice threshold based by tuning the model output to observations, when a model-observation comparison is being performed. In the end, it doesn't really matter because the value is more or less the same as the value of 10 cm noted in the citation for the observational product, but I don't really understand why the authors didn't just use the 10 cm value.

The reason is that the 9.8 cm value is the result of analysing the model results, while the 10 cm value is very approximate and probably not uniform across space and time (variations in surface and atmospheric conditions may affect it, for instance). We therefore us 9.8 cm, arguing that using the parameter that results from analysing the model results is the appropriate choice for the comparison, rather than using the approximate value given by the observationalists.

The authors have added some broader motivation in the Introduction, but I think the manuscript would still benefit from some discussion of broader implications in the Conclusions/Summary (which is rather abrupt).

We have now added a concluding paragraph to the "summary and conclusions" section to better address this comment.

Line-by-line comments (should be addressed)

L21: remove 'in the centre of the Arctic Basin' after 'Leads'

Done

L40: When referring to 'larger' and 'smaller' leads, what is the approximate length scale?

This was more-or-less a direct reference to Marcq and Weiss (2012), but on re-reading we found it to be difficult to do it justice in a concise manner; the way they end up with the 55% figure is a bit involved. We therefore replaced ", in that particular case, the sensible heat flux over the smaller leads could dominate that over the larger ones, such that including smaller 40 leads increased the total estimated heat flux by 55%" with "turbulent heat transfers between the ocean and the atmosphere in ice-covered oceans strongly depends on the distribution of lead widths, especially at very small scales (smaller than 50 m)".

L45: 'model results are lacking in several aspects.' Such as?

Again, we found it difficult to be sufficiently concrete here, as there is substantial difference between models, and it is not yet clear what causes this. We therefore decided to remove this sentence part.

L58: 'he' should be 'the'

Fixed

L62: 'the statistical properties of leads in large-scale sea-ice models have not yet been shown to be robustly reproduced' I think this means to say that the models have not yet been shown to reproduce observations

Indeed. We have rephrased this to make it clearer.

L64: Add comma after deforms

Done

L110: Typo, 'comparable'

Fixed

L115: Typo, 'results'

Fixed

L124: You might add that the cohesion parameter appears in the Mohr-Coulomb criterion

Done

L126: Are the Bouillon & Rampal 2015a and 2015b references listed the right way round? I think they might be mixed up

Yes, they were. We've fixed that now.

L169: Mentions snapshots – but text above states that only daily means are used. Remove 'snapshots'?

Done, this got left over from an earlier version of the manuscript.

L178: Perhaps say, 'as a a function of spatial scale'

Yes – changed.

L179: Remove 'the', before 'log-log space'

Done

L186: Suggest removing the first part of the sentence, and simply say 'B(q) is the structure function, which '…

Done

L190: This paragraph is a bit clumsy, consider rewording

Done

Fig. 1 caption: The last sentence should be in the main text rather than in the caption. How do you know that the observational product displays 'noise' rather than real leads? This needs to be justified.

We've removed this, as discussed above.

L208: suggest rewording to 'Figure 2 shows the PDF of observed and simulated lead fraction in the Central Arctic on a log-log scale. On this scale, both the observed and the simulated PDF'…

Done

Fig. 3 caption: should include brief description of the error bars

Done

L271: I don't think differences of 20% and 30% should be described as 'nearly the same'

Indeed not, this was a mistake. We've changed "nearly the same" to similar.

L292: 'This fractal character is inherited from the multifractal character of the ice deformation rates, as discussed below.' This sentence is unnecessary and can be removed.

Done

L369: replace 'should otherwise' with 'would otherwise'

Fixed

**On the statistical properties of sea ice lead fraction and heat fluxes in the Arctic**

Einar Ólason[1], Pierre Rampal[1,2], and Véronique Dansereau[1,3]

[1]Nansen Environmental Remote Sensing Center and Bjerknes Centre for Climate Research, Bergen, Norway
[2]Now at Université Grenoble Alpes, CNRS, Grenoble INP, Institut de Géophysique de l'Environnement, Grenoble, France
[3]Now at Université Grenoble Alpes, CNRS, Grenoble INP, Laboratoire 3SR, Grenoble, France

**Correspondence:** Einar Ólason (einar.olason@nersc.no)

**Abstract.**  We explore several statistical properties of the observed and simulated Arctic sea-ice lead-fraction, as well as the statistics of simulated Arctic ocean-atmosphere heat fluxes. First we show that the observed lead fraction in the Central Arctic has a monofractal spatial scaling, which we relate to the multifractal spatial scaling present in sea-ice deformation-rates. We then show that the relevant statistics of the observed lead fraction in the Central Arctic are both well represented by our model, neXtSIM. Given that the heat flux through leads may be up to two orders of magnitude larger than that through unbroken ice we then explore the statistical properties (PDF and spatial scaling) of the heat fluxes simulated by neXtSIM. We demonstrate that the modelled heat fluxes present a multifractal scaling in the Central Arctic, where heat fluxes through leads dominate the high-flux tail of the PDF. This multifractal character relates to the multi- and monofractal character of deformation rates and lead fraction. In the wider Arctic, the high-flux tail of the PDF is dominated by an exponential decay, which we attribute to the presence of coastal polynyas. Finally, we show that the scaling of simulated lead fraction and heat fluxes depend weakly on the model resolution and discuss the role sub-grid scale parameterisations of the ice heterogeneity may have in improving this result.

*Copyright statement.* The author's copyright for this publication is transferred to the Nansen Environmental Remote Sensing Center, Bergen, Norway.

[revised manuscript text omitted]

---

## Author Response (AR3)

Dear editor,

Thank you very much for your constructive comments. We have addressed all of them and we hope that our modifications are to your satisfaction.

Below we have copied out your comments in black and inserted our response in blue type.

Regards,

Einar (on behalf of the authors)

We expect you mean for us to change to "0.1 m as thin ice". However, cm is an SI unit and using cm in this case infers a sense of scale; using cm indicates that the physical properties considered are on the range of centimetres, not metres or kilometres. Indeed, the resolution of the model and observations is quoted in kilometres, not metres, for this very reason. We have, therefore, not changed this.

Comments to the Author:
I thank the author for their rewrite of this manuscript which resulted in a much approved version. Before final acceptance, pls provide an updated version with view of the technical issues identified below.

General:
* I abbreviate "manuscript" with "ms".
* Throughout the ms, pls correct "e.g." to "e.g.,".

Done

* Ensure use of SI units throughout the ms, e.g., 136 "10 cm" -> "0.1m".

We have changed this, seeing you insist on it. However, cm is an SI unit and using cm in this case infers a sense of scale; using cm shows that the physical properties considered are on the range of centimetres, not metres.

* With advise from the TC Editorial Team, pls apply correct reference style for figure and table calls. I.e., 225 "(figure 4b)" does not look correct.

We have changed all references to figures to read "Fig.", instead of "figure". The example on line 225 then reads "(Fig. 4b)".

Specific comments:
67: Change "give us" to "provide".

Done

68: Corret "solids as sea ice" to "solids such as sea ice".

Done

70: Change "which in turn triggers more deformation nearby" to "which in turn may give rise to further ice deformation nearby".

Done

72: Capitalize "Earth".

Done

73: Correct "smaller quakes more distributed in the vicinity" to "smaller quakes occurring in the wider vicinity".

Done

75: Change "using observations and the neXtSIM model" to "using both, observations and the neXtSIM model".

Done

76: Correct "we present briefly" to "we briefly present".

Done

77: Correct "In section 3" to "In Section 3".

Done

80: Change "— which is done in section 4" to "(Section 4)".

Done

81: Correct "section 5" to "Section 5".

Done

82: Correct "depends" to "depend".

Done

82: Correct "section 6" to "Section 6".

Done

84: Correct "Arctic basin." to "Arctic Basin."

Done

91: Correct "thermodynamics model" to "thermodynamic model".

Done

94: Remove "a category of".

Done

122: Suggest to replace "to degrade significantly" with "to diverge significantly".

Done

136: Pls use SI units.

Done

145: Clarify the exact mathematical procedure when "combine all the daily means", i.e., explain "combine".

The pdf is calculated from all daily means in within the relevant regions during the period JFM, 2007. We've rephrased the sentence to read "The PDFs shown in this paper are calculated from all the daily means for JFM, 2007, within the regions outlined below (Section 3), for both the lead fraction and heat flux magnitudes."

182: Pls use SI units.

Done

194: Rewrite "the PDF for the observations flattens out" to improve English.

We've rewritten this sentence to read "For values smaller than about 40%, the slope of the PDF of the observations changes to approach zero for very small values."

Caption Fig. 5 right: I cannot see a line for "linear ... fit".

The caption was incorrectly formulated. We've changed it to read "... Right pane: The resulting structure function and a quadratic fit. The error bars are the uncertainty of the least-squares fit for the moments (left pane)."

Caption Fig. 5: Correct "topp" to "top".

Done

267: Correct "12 km" to "12.5 km".

Done

327: Correct "in at" in "smaller effect in at the time scales".

Done

376: Correct "this data" to "these data".

Done

376: Correct "The source code to neXtSIM" to "The neXtSIM source code".

Done

Acknowledgement: Suggest to change "an anonymous reviewer" to "an anonymous 2nd reviewer".

Done